# The Mental Health Benefits of Purposeful Activities in Public Green Spaces in Urban and Semi-Urban Neighbourhoods: A Mixed-Methods Pilot and Proof of Concept Study

**DOI:** 10.3390/ijerph16152712

**Published:** 2019-07-30

**Authors:** Peter A. Coventry, Chris Neale, Alison Dyke, Rachel Pateman, Steve Cinderby

**Affiliations:** 1Department of Health Sciences and Centre for Reviews and Dissemination, University of York, York YO10 5DD, UK; 2Frank Batten School of Leadership and Public Policy, University of Virginia, Charlottesville, VA 22904, USA; 3Stockholm Environment Institute, Department of Environment and Geography, University of York, York YO10 5DD, UK

**Keywords:** green space, mood, well-being, activity, location, urban

## Abstract

Access and exposure to public green space might be critical to health promotion and prevention of mental ill health. However, it is uncertain if differential health and mental health benefits are associated with undertaking different activities in public green space. We evaluated the health and wellbeing benefits of different activities in different locations of public green spaces in urban and semi-urban areas. We used a mixed-methods before-and-after design. Volunteers at three conservation sites were recruited and took part in group guided walks, practical conservation tasks or citizen science. Repeated measures one-way ANOVAs with Bonferroni correction assessed the relationship between location and activity type on change in acute subjective mood from pre- to post-activity, measured with the UWIST Mood Adjective Checklist (UWIST-MACL). Qualitative semi-structured interviews were undertaken and analysed thematically to explore participants’ perceptions about the health and wellbeing benefits of activities in public green space. Forty-five participants were recruited, leading to 65 independent observations. Walking, conservation and citizen science in public green space were associated with improved mood. Across all participants acute subjective mood improved across all domains of the UWIST-MACL. There was a significant association between reduction in stress and location (*p* = 0.009). Qualitatively participants reported that conservation and citizen science conferred co-benefits to the environment and individual health and well-being and were perceived as purposeful. Undertaking purposeful activity in public green space has the potential to promote health and prevent mental ill health.

## 1. Introduction

By 2050, 68% of the global population will live in cities, heralding an unprecedented era of urbanisation [1]. Urban living is associated with exposure to environmental stressors arising from high levels of air, noise and heat pollution which are associated with higher odds of developing common mental health problems such as depression and anxiety [2]. Additionally, urban up-bringing is associated with a two-fold increased risk of psychotic disorders in adulthood [3]. Population density and urban design also contribute to social isolation which has been implicated in poorer mental and physical health [4,5]. 

Depressive disorders are the leading global cause of disease burden accounting for most disability-adjusted life years across developed and developing regions [6]. In the UK 1 in 4 people will experience a mental health problem. Furthermore, the combination of long-term physical conditions and depression is associated with the greatest decrements in quality of life and is associated with poorer outcome of long-term conditions, with significant cost implications [7,8].

There is now an emerging policy consensus that novel prevention strategies are critical to reengineering the way we improve mental health and promote wellbeing and tackle isolation and loneliness [9,10]. This is likely to be achieved by developing population health strategies that blend intelligence about the wider determinants of ill-health with the development and implementation of placed-based solutions, with a focus on improving mental and physical health together [11]. Improved access to public green space and promotion of outdoor activity and social interaction are among key potential solutions to the challenges of population health in cities in the 21st century [12]. Global standards now exist for sustainable urban development that map to achieving the Sustainable Development Goals (SDGs) which include the aim, by 2030, of providing universal access to safe, inclusive and accessible public green spaces, particularly for women and children, older people and people with disabilities [13]. The UK Government’s 25 Year Environment Plan signals a shift towards recognising the importance of urban green infrastructure for delivery of the SDGs [14].

Residential greenness is consistently associated with lower odds of depression and the quantity of green space in the living environment is positively associated with perceived mental health and general health too [15,16]. Exposure to residential green space during childhood is also linked with a lower risk of developing psychiatric illness in adolescence and adulthood [17]. Furthermore, the provision of parks within walking distance, including those with a recreational and sporting focus are likely to be important for positive mental health [18]. However, it is not well known whether different types of activities in green space are associated with differential effects on mental health [19]. Population level analysis has shown that adults over 45 years have lower levels of distress if they are physically active in greener surroundings, but exposure to green space among those who do no or little activity has no impact on mental health [20]. Additionally, in-situ neuroimaging has shown that short walks in urban green spaces reduce neural activity associated with stress and increase activity associated with relaxed states [21]. It might be, therefore, that mental health benefits are dependent on physical activity in urban green space. Additionally, it is unclear if other forms of green space activity that offer cerebral engagement and social interaction are important drivers of health and wellbeing outcomes.

There is a need to undertake experimental and quasi-experimental work that investigates the before-and-after effects of exposure to different types of activities across different types of public green space. In addition to measuring exposure to public green space it is also important to explore people’s motivations for using green space and their perceptions about the benefits to gain insight about how to improve and sustain accessibility to green space [22]. We therefore conducted a before-and-after mixed-methods pilot study to test the feasibility of evaluating the mental health benefits among conservation group volunteers of different types of outdoor activity in three different public green spaces in urban and semi-urban areas. We also qualitatively explored participants motivations for taking part and continuing activities in public green space to better inform the design of future experimental studies about the mental and physical health benefits of public green space.

## 2. Methods

### 2.1. Identification and Recruitment of Participants

Participants were conservation volunteers who attended green spaces managed by the Yorkshire Wildlife Trust; The Conservation Volunteers; and the Friends of St Nicholas Fields (St Nicks), York. Two members of the research team (CN; RP) emailed copies of the participant information sheet to volunteer group leaders who forwarded it to potential participants before they attended a volunteer session at the site they would normally attend. Researchers gave participants the option of taking part, and those who accepted were given the opportunity to ask any questions or raise any concerns they might have before giving written informed consent.

### 2.2. Experimental Sites

In urban and semi-urban contexts public green spaces can be accessed freely by all citizens and are conceived of as natural or semi-natural environments that are partially or completely covered by vegetation, (e.g., parks, playing fields, public gardens, forests, woods and nature reserves) as well as human-modified places (e.g., riverside greenbelts, institutional green spaces, green squares) [23,24]. Testing was undertaken at three different public green spaces across York, England and the surrounding area.

Site 1: Askham Bog (OS Map Reference SE574479) is a green space on the outskirts of York made up of a mosaic of fenland, meadow and woodland. It is a Site of Special Scientific Interest with boardwalks and walking trails throughout and is managed by the Yorkshire Wildlife Trust (Figure 1).

Site 2: St Nicks (OS Map Reference SE6169851812) is a community green space on a former landfill site in a residential urban area of York, now a mix of grassland and woodland. It is managed by the Friends of St Nicholas Fields who offer a suite of ‘ecotherapy’ activities specifically aimed at promoting good physical and mental wellbeing (Figure 2).

Site 3: A large green field with surrounding woodland adjacent to a semi-urban housing development in Barlby (OS Map Reference SE6306133664). Habitat management activities are undertaken by The Conservation Volunteers (Figure 3).

### 2.3. Activities

The protective and health beneficial effects of urban green space have been attributed to one or more of four putative mechanisms of action: (1) By restorative exposure that reduces attention fatigue and stress [25,26]; (2) By facilitating social interaction and enhancing social cohesion [27]; (3) By facilitating and promoting physical activity [28]; (4) By filtering out the negative effects of noise, air and heat pollution [29]. We proposed to test the mental health benefits of activities that addressed at least three of these mechanisms: restoration, social interaction, and physical activity.

Group walks (Askham Bog and St Nicks): participants undertook directed out-and-back walks in small groups. While walking participants were asked to keep from engaging in conversation with other group members so as to maximise focus on the physical activity elements of walking. This activity was designed and implemented as the most passive and least interactive activity to offer a comparison with the other two activities.

Citizen Science (Askham Bog and St Nicks): participants took part in the OPAL Air Survey to assess impact of local air quality on the environment by studying lichens found on trees, which are indicative of nitrogenous pollution [30] or the OPAL Tree Health Survey to look for signs of pests and diseases on trees. Results are uploaded to the OPAL survey website. This activity was designed to be socially interactive and restorative.

Conservation (Barlby; Askham Bog; St Nicks): participants undertook practical conservation tasks such as flood mitigation, scything, pruning and creating wildlife habitats. This task was designed to be interactive and restorative and potentially included physical activity.

The activities and evaluations all took place between September and October 2017 and the weather was characterized by warm and sunny intervals with no rain.

### 2.4. Outcomes

#### 2.4.1. Subjective Wellbeing

Measured by the Short Warwick–Edinburgh Mental-Wellbeing Scale (SWEMWBS). This is a measure of self-reported wellbeing taken from personal experiences over the last two weeks. It is has been validated as a reliable measure of wellbeing in adult populations as well as being used in environmental research [31,32]. The output from this measure gives a single ‘wellbeing’ score where a higher score indicates a higher level of self-reported wellbeing. These scores are obtained from a summation of scores from the responses to each of the items within the scale.

#### 2.4.2. Acute Subjective Mood

Measured by the University of Wales Institute of Science and Technology (UWIST) Mood Adjective Checklist (MACL). This is a checklist used to determine acute subjective mood changes [33]. The UWIST MACL is a 24-item checklist that gives acute measures of hedonic tone (valence), stress and (physical) arousal, shown as three scores. Respondents are required to complete the questionnaire before and soon after completion of activity to ensure measurement of momentary shifts in mood. The arousal scale measures feelings of subjective energy. The stress scale measures feelings of subjective tension and the hedonic tone scale measures overall pleasantness of mood, and is associated with feelings of somatic comfort and wellbeing. Scores are obtained from summation of individual item scores pertaining to each of the three mood components.

### 2.5. Before-and-After Design Procedures

At baseline, participants were are asked to fill out a short demographic questionnaire as well as the SWEMWBS and UWIST questionnaires. All activities lasted between 20 and 30 min. Participants at Barlby and St Nicks undertook activities on one occasion only. Participants at Askham Bog were able to take part in more than one activity across three sessions. completion participants completed a second and final UWIST, concluding their experimental participation. Semi-structured interviews were undertaken with a sample of participants 8 weeks after completion of activities. The topic guide is shown in online Appendix A and focused on exploring participants’ motivations for starting and continuing environmental volunteering and their perceptions about the perceived benefits of undertaking activities in green space.

### 2.6. Data Analysis

#### 2.6.1. Quantitative Analysis

Both UWIST MACL and SWEMWBS scales were completed on paper and scored by the research team for each of the relevant time points during the experiment. Demographics and other baseline data were collected by self-report. Between group comparisons were made using a repeated measures one way ANOVA that explored each of the three mood outcomes by both location and activity type. In all cases, a post-hoc pairwise Bonferonni correction for multiple comparisons was used to explore any overall significant effects of each model. All statistical tests were performed with IBM SPSS Statistics 25 software (IBM Corporation, Armonk, NY, USA) and the significance level was set at α = 0.05. The quantitative data set is available in Appendix A.

#### 2.6.2. Qualitative Analysis

Interviews were audio recorded and listened to by one researcher who populated an initial coding frame in Excel (AD). Analysis was guided by the principles and procedures of the thematic analysis [34]. There was inductive initial coding, followed by re-coding and memo writing in order to generate conceptual themes across interview topics related to motivations for environmental volunteering and perceptions of benefit of outdoor activity. Consensus meetings between the study team then discussed and agreed overarching interpretations. The qualitative audio recordings are available on request from the corresponding author.

Ethical approval for the study was provided by the University of York Department of Environment and Geography Ethics Committee.

## 3. Results

### 3.1. Sample Characteristics

Forty-five participants were included in the study; nine participants at Askham Bog undertook multiple activities leading to 65 independent observations across the three sites. Characteristics of participants are shown in Table 1. Our total sample had a higher wellbeing score than the population norm for England. Scores among the Askham Bog volunteers (25.29, (standard deviation [SD] = 4.8) and in Barlby (30, SD = 4.15) were higher than the population norm for England (23.6, SD = 3.9), whereas the volunteers at St Nicks had level of wellbeing lower than the national average (21.79, SD = 4.59) [35]. Three people at St Nicks reported that they had severe mental illness and two people at Askham Bog reported they had common mental health problems (generalised anxiety disorder and depression).

### 3.2. Mood

There was an overall positive change in hedonic tone from pre- to post-activity across all participants at all locations (mean difference = 2.13; 95 % confidence interval (CI) = 0.78 to 3.45, *p* = 0.003). However, there was no significant difference in pre-to post-activity scores for hedonic tone by location (Figure 4).

When assessing whether hedonic tone was differentially affected by type of activity there was an overall positive change in pre- to post scores across all participants and all activities (mean difference = 2.44; 95% CI = 1.34 to 3.54, *p* ≤ 0.001), but no significant difference for this effect by type of activity.

There was an overall positive effect on pre- to post-stress scores across all participants at all locations (mean difference = −3.53, 95% CI = −4.79 to −2.28, *p* ≤ 0.001). The Bonferroni Correction shows that differences between pre- and post-stress levels are most significant between Askham Bog and St Nicks (mean difference = −5.05; 95% CI = −7.92 to −2.18, *p* < 0.001), with further significant effects between St Nicks and Barlby (mean difference = −5.17; 95% CI = −9.68 to −0.66, *p* = 0.020). This suggests that the magnitude of the reduction in stress shown in Figure 5 was most pronounced in the St Nicks group.

When assessing whether stress was differentially affected by type of activity we found that there was an overall positive effect on pre- to post-activity scores across all participants (mean difference = −3.60; 95% CI = −4.69 to −2.51, *p* ≤ 0.001), but no significant difference for this effect by type of activity.

There was an overall positive change in arousal from pre- to post-activity across all participants at all locations (mean difference = 2.90; 95% CI = 1.07 to 4.75, *p* = 0.03), but no significant difference for this effect by location.

When assessing whether arousal was differentially affected by type of activity we found that there was an overall positive effect on pre- to post-activity scores across all participants (mean difference = 3.61; 95% CI = 2.20 to 5.02, *p* ≤ 0.001), but no significant difference for this effect by type of activity (Figure 6).

### 3.3. Qualitative Findings

We interviewed six volunteer participants (five from Askham Bog and one from Barlby) and two members of staff (one from St Nicks and one from the Yorkshire Wildlife Trust) responsible for running volunteering at the green space sites. Five volunteers were male, one was female and all were retired; both staff members were female. We have not presented other demographic details about interviewees to ensure anonymity is preserved.

#### 3.3.1. Predisposition and Activation

In explaining motivations for why they engaged in environmental volunteering, many of the participants revealed that they had a historical predisposition to spending time being active outdoors. For some being outdoors was an intrinsic feature of their life:
“I’ve always liked spending time outdoors and being active”.(TAP003)
“It’s essential to me, it always has been”.(TAP042)

When asked to identify what types of green spaces they preferred, one participant suggested they liked places that were functional and were characteristically working landscapes. However, the majority expressed a preference for green spaces that were akin to wilderness, or at least were less managed:
“I like wild spaces where there has been less obvious or lower levels of human management”(TAP 052)

In-keeping with preferences for wilderness many participants also favoured green and blue spaces “…where there aren’t too many people” (TAP003) and social contact was minimal:
“I like the peace of the beach where human contact is less and where there is space and solitude” (TAP 029)

Pragmatically however many of the participants recognised that urban public green spaces offer opportunities to experience some of the qualities of wilder green spaces and working in more managed green spaces sustained their lifelong connection with being outdoors.

In addition to having a shared sense of affinity with natural environments, participants in this study were activated to engage in environmental volunteering because it offered a way to structure their retirement:
“I went down almost the first Wednesday after I finished work”.(TAP022)

Or the prospect of environmental volunteering offered opportunities to take part in activities that differed markedly from office bound and sedentary jobs. Here there was a sense that outdoor activity was likely to be beneficial to maintain their physical health, but equally others signalled that joining an environmental volunteering group was key to maintaining their mental health:
“Part of it is physical, but also mental wellbeing. I wasn’t even going to bump into former colleagues, so I had to find new ways of meeting people and ways to get out of the house”.(TAP043)

#### 3.3.2. Motivations for Continuing

Many of the participants continued to take part in environmental volunteering because they perceived it to be a purposeful and meaningful activity. For some, the purpose was articulated in how the activity was analogous to work itself, which was a driving force throughout their life:
“I wanted to do some work outside, because I’ve found that work is also necessary to me even if I don’t get paid for it”.(TAP042)

Also fundamental to continued participation was the chance to dedicate themselves to improving the natural environment:
“I want to see the reserve being maintained properly”.(TAP003)
“You can see the changes that we have made in some reserves. Some of the reserves we have made a real difference to”.(TAP022)

Participants recognised that the utility of working with a conservation group and improving the environment offered them scope to not only be physically active but to derive satisfaction from purposeful work:
“I felt I’d had good activity and I could see I was doing something worthwhile in an important place—so it ticked the boxes of green space that was doing something useful”.(TAP022)

For many participants their on-going relationship with the natural environment was mediated by the conservation group leaders who were able to contextualise how volunteering contributed to nature conservation:
“…they [group leaders] are very good at explaining why we are doing all this cutting and raking”.(TAP043)

By being educated about the philosophy and goals of conservation participants developed loyalty to particular groups and prompted them to seek out more environmental volunteering opportunities within those groups:
“I’m now more interested in the cause of nature conservation”.(TAP042)

Additionally, for some of those interviewed, continued participation with their conservation group led to stronger social bonds with fellow volunteers:
“I appreciate the social aspect more”.(TAP003)

Often group leaders played a key role in facilitating access to the groups and making people feel welcome:
“We got on, I was looked after, was greeted and welcomed by the group leader”.(TAP022)

And for some participants the group leaders were instrumental in maintaining their motivation to contribute to the group’s activities:
“…they [group leaders] are what would get me out on a miserable winter day as well as on a lovely summer day”.(TAP052)

While this sense of camaraderie and strong identity with the group’s mission enabled some participants to feel part of a tight-knit community they recognised that for others, not accustomed to the group, it could make environmental volunteering difficult:
“You can tell if someone’s having a bit of a career break and they are trying to get themselves set up again, it can be hard to fit in”.(TAP043)

#### 3.3.3. Health and Mental Health Benefits of Green Space Activities

In thinking especially about the physical and mental health benefits of the different activities the interview data confirmed findings from the quantitative analysis with general declarations about how being engaged in the various activities gave participants satisfaction which was reflected in improvements in mood across all the locations. Participants highlighted that conservation work was physically demanding and they used muscle groups they would not normally use. However, many of the conservation tasks could be repetitive, boring or uncomfortable, and participants were content to switch tasks or enjoy social companionship instead:
“A few times when I’ve not been happy or not been able to continue, you can do something else. I can’t hammer and bang for very long because of my wrists, so you do something else like pull Himalayan balsam in the woods or chat”.(TAP 042)

As much as the conservation work was activity directed by the group leaders, participants felt they had some control over how it was undertaken, giving them a sense of agency and greater satisfaction with their work:
“We were rolling up netting on the old boardwalk prior to demolishing it. It was very difficult in the way that we were supposed to do it, so we ended up just smashing it all up. A good thing is that you are at liberty if you don’t like the way that you have been told to do something to do it a different way if you can think of something better”.(TAP 042)

When given opportunities to take part in citizen science some participants were sceptical about its merits and were concerned that it would delay their involvement in their usual conservation activity. However, for those that were not always drawn to the physical aspects of conservation work, citizen science offered a chance to enjoy being outdoors in novel and unusual ways:
“…[I] had turned up expecting to work on the boardwalk. I’m not skilled so try to avoid construction. So on boardwalk days that tends to involve carting wood and I might not have been as enthusiastic as otherwise. I was a bit puzzled initially…[but] was surprised to feel better afterwards”.(TAP022)

Learning and discovery were key features of citizen science that attracted participants to this activity. Participants who took part in the lichen survey acquired new knowledge about an unfamiliar topic and they also gained new insights into their green space through observing them in novel ways.

Similarly, discovery featured in participants’ descriptions of group walks which gave them a chance to visit parts of the reserve that they would not normally have access to. In addition, despite being encouraged to walk by themselves some participants found that walking offered opportunities to interact with others which conferred social benefits. Walking also offered moments to enjoy solitude and for participants to take their own path, and to be closer to wildlife. One group leader recalled purposively taking a longer route and embraced a mindful moment to “…concentrate on the birds, to hear the birds, the weather, [and] what had changed” (TAP29).

### 3.4. Willingness to Be Randomised

In thinking about planning future randomised evaluations of the health benefits of activity in green space opinion was split between those participants who would not mind being allocated to a different activity and those who would want to continue with their conservation work alone or at least be given a choice. Among those who might want to avoid taking part in a randomised evaluation were those who suggested they could be persuaded to do so if the rationale was explained and the benefits to science were clear.

## 4. Discussion

We showed that we can recruit and retain conservation group volunteers to measure before-and-after effects of different activities on subjective mood in urban and semi-urban public green spaces. Additionally, as proof of concept that mental health benefits are derived from purposeful activities in green space, we showed that citizen science and conservation were associated with improvements in subjective mood. Magnitude of change in stress changed with location of green space but this was not the case for hedonic tone or arousal. Qualitative exploration showed that outdoor activities in urban and semi-urban green public space structured daily life in retirement and offered opportunities to be exposed to natural environments in the absence of exposure to wilderness. While walking was seen as health beneficial, citizen science and conservation were characterised as purposeful and meaningful activities that yielded co-benefits for individual health and wellbeing and also for green spaces. Citizen science and conservation also offered opportunities for learning and discovery. Citizen science and conservation activities enhanced social interaction, in part to counter balance more isolated existences away from the world of work, but also to generate a sense belonging among fellow volunteers who shared aspirations to learn more about their local green space. Participants also expressed a general willingness to take part in a randomised evaluation if the process was justified, with some inclined to only take part in a study where they could state a preference for a particular type of activity.

Positive effects on mood were observed for all activities but there was a signal that the location of the green space, over and above the type of activity, was an important factor in reducing stress. The concept of a healthy place is not only dependent on the physical characteristics of a site but also includes a sense of place that people attach to it and this is accrued over time through lived experience with that place [36]. Our qualitative findings suggest that the volunteers who attended St Nicks were especially loyal to that organisation and had historical ties with the green spaces that group managed, suggesting that place attachment and place identity are important factors in driving the relationship between exposure to natural environments, activity and wellbeing.

Conservation was the activity that participants had the most affinity with owing to their previous experiences with the conservation groups. Taking part in conservation activities improved hedonic aspects of wellbeing related to reductions in stress and increased vigour and energy, but our qualitative findings also suggest these activities improved eudaimonic wellbeing too. Eudaimonic wellbeing is associated with self-actualisation and living a full and purposeful life [37], and these attributes map to the finding that conservation work was perceived as purposeful and meaningful with co-benefits for individuals and the environment. Husk et al. similarly found that people who took part in environmental enhancement and conservation activities had higher levels of perceived health owing to a range factors that included a sense of achievement [38]. Likewise, citizen science was perceived as purposeful and broadened people’s understanding of their green space, and in this sense may also have improved eudaimonic as well as hedonic wellbeing. To date most work on place-based citizen science has pointed to its role in raising environmental awareness and changing behaviour [39], but we have shown that it also has the potential to confer individual and wellbeing benefits.

Walking also improved acute subjective mood. While this was the most passive activity that we assessed, we know that taking a walk or a run in natural environments is better for wellbeing than comparable activity in synthetic environments [40]. Short walks of thirty minutes in natural environments with green or blue spaces are also associated with greater restorative experiences and cognitive function than similar walks in urban environments [41].

There is also further confirmatory evidence that participants in this study experienced improvements in mood through the restorative experience of being in nature, lending some support to attention restoration theory. Attention restoration theory proposes that our ability to concentrate on a task requires directed attention which is finite and more likely to be used up in urban and high stress environments [42]. Attention fatigue has been implicated in poorer decision making, stress and less self-control, leading to physical and mental ill health. Time spent ‘being away’ in natural environments that afford opportunities to engage in activities that are ‘softly fascinating’ and are compatible with individual needs can bring about involuntary or effortless attention and restore our directed attention capacities [43]. Conservation work might especially be softly fascinating in that it is typically goal-directed, repetitive, ordered yet creative: enabling the participants to enter what is referred to as an immersive “flow state” situated somewhere between skill and challenge. A flow state is akin to a mindful state. By cultivating a focus on the present, rather than ruminating on the past or worrying about the future, mindfulness states can enable people to more effectively cope with physical or mental stressors that can negatively affect wellbeing. Citizen science and to a lesser extent walking may also have offered participants opportunities to engage in such mindful states and thereby become less stressed. In particular the scale of the organisms that participants were asked to observe in the citizen science activity might have heightened a sense of soft fascination. Lichens, for example, are small and focusing on these to the exclusion of other factors could have lent participants greater capacity to mindfully engage in this task.

### 4.1. Strengths and Limitations

All the participants were existing volunteers for conservation and wildlife groups located in an affluent part of North Yorkshire, and in that sense our study is prone to selection bias. Participants were more likely to have been engaged in the activities and inclined to give positive responses about exposure to nature owing to their previous relationship with the volunteer groups. However, we offered participants opportunities to take part in activities that they were not accustomed to and no one dropped out showing that we can feasibly conduct quantitative and qualitative assessments across multiple green space sites. Because this was a pilot study the sample size was small, and we were not able to definitively measure the impact of activities on mood. We only measured the before-and-after effects of activities in green space on hedonic aspects of wellbeing but our qualitative evaluation captured data that relates to eudaimonic wellbeing which may have an important role to play in preventing mental ill health. Additionally, we did not assess if other characteristics moderated the effect of activity and location on wellbeing. Only a small sub-set of participants were interviewed limiting our ability to reach data saturation and derive themes with any certainty. However, the interviews contributed to evaluation of feasibility and proof of concept of conducting a before-and-after study of the impact of activities in urban and semi-urban green space. In this sense the interview data produced solid data to inform future generation of research hypotheses and provided intelligence about optimal designs of future studies of exposure to urban and semi-urban green space and wellbeing.

### 4.2. Future Research

Given the potential to modify the use of public green space in urban centre to promote health and prevent mental ill health there is a need to further investigate the most effective ways to harness the health and wellbeing capital of cities [44]. We showed that it is likely to be possible to conduct randomised evaluations of activities in green spaces and future studies should include measures of eudaimonic as well as hedonic wellbeing. Critically, to isolate active ingredients in green space interventions it would be advantageous for future studies to include a control group that exposed participants to a non-green space activity that acted as an attention control. Additionally, such future studies should also include process evaluations that aim to explain the mechanisms of action with reference to factors that might mediate and moderate the relationship between activities in and connection with nature and wellbeing outcomes [45,46]. Children are especially vulnerable to the impact of urban related stressors such as air and noise pollution [47]. Exposure to green space may be protective for children at risk of schizophrenia in adulthood [48]. Older adults are similarly vulnerable to urban related stressors and there is emerging evidence that engaging in purposeful activity is protective against weak grip strength and decline in walking speed and as such can delay or slow the onset of frailty [49]. There is a priority to test the effectiveness of purposeful activity in public green space to promote and preserve physical functioning and prevent mental ill health among older and socially isolated adults. Further, while the positive effects we observed on mood stemmed from undertaking activities in green space over a relatively short period we did not measure dose-response. There is some evidence that health and well-being benefits can occur soon after exposure to nature interventions and that prolonged exposure does not yield further benefit [50,51]. This has important implications for delivery and scaling of green space interventions in cities through such platforms as social prescribing, especially for use by children and socially isolated older adults.

## 5. Conclusions

Taking part in conservation, citizen science or walking in public green spaces in urban and semi-urban neighbourhoods is associated with positive effects on mood. Undertaking such activities in locations where people have the most connection might confer additional benefits. Those activities that were characterised as purposeful and meaningful were associated with the most perceived physical and mental health benefits. Social interaction, physical activity and restoration were all implicated as potential mechanisms by which activities in public green spaces might lead to improved mental health. There is scope for a randomised evaluation of the mental health benefits of activities in green spaces in cities that target populations at high risk of mental ill health and frailty.

## Figures and Tables

**Figure 1 ijerph-16-02712-f001:**
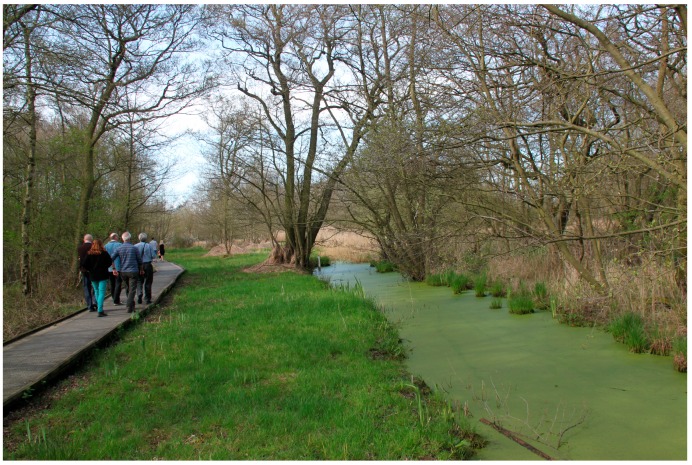
Askham Bog Nature Reserve, Yorkshire Wildlife Trust.

**Figure 2 ijerph-16-02712-f002:**
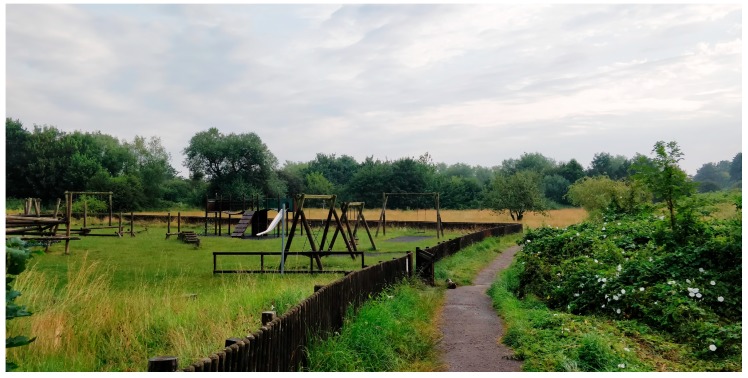
St Nicks Nature Reserve, York.

**Figure 3 ijerph-16-02712-f003:**
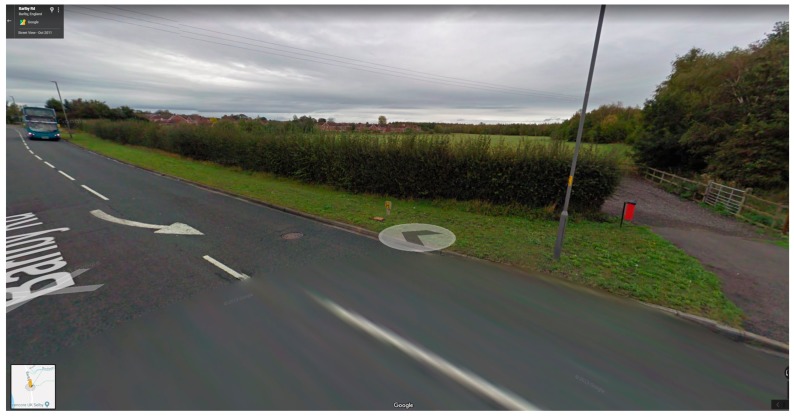
Barlby greenfield site, Barlby Road, Yorkshire.

**Figure 4 ijerph-16-02712-f004:**
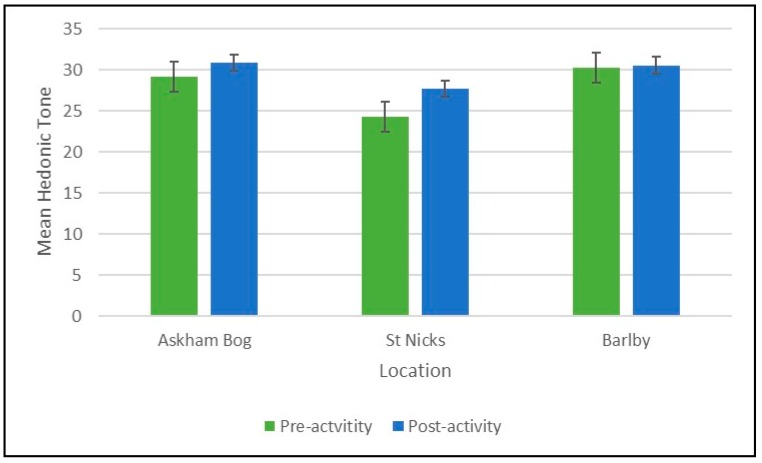
Mean pre- and post-hedonic tone scores by location.

**Figure 5 ijerph-16-02712-f005:**
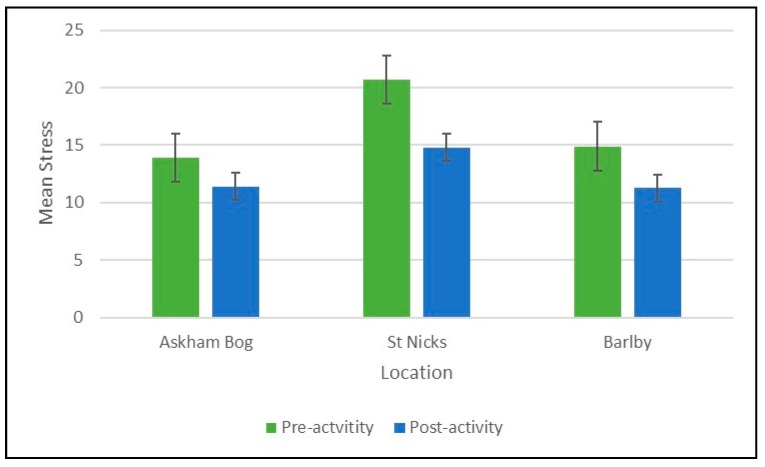
Mean pre- and post-stress scores by location.

**Figure 6 ijerph-16-02712-f006:**
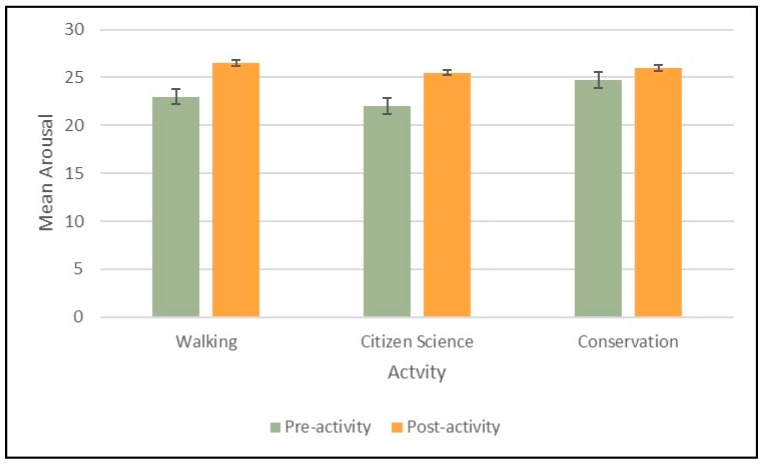
Mean pre and post arousal scores by activity.

**Table 1 ijerph-16-02712-t001:** Characteristics of participants.

Characteristics	*n* = 45
Age (mean, SD)	43.8 (SD 2.83)
Male (*n*, %)	25 (59%)
SWEMWBS (mean, SD)	24.9 (5.1)
Self-reported mental health problems	
Asthma	1
High blood pressure	1
Diabetes	1
Anxiety	2
Depression	2
Autism spectrum disorder	1
Schizophrenia	2
Bipolar	1

SD = standard deviation; SWEMWBS = Short Warwick–Edinburgh Mental Wellbeing Scale.

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
