# Peer review of "The Mental Health Benefits of Purposeful Activities in Public Green Spaces in Urban and Semi-Urban Neighbourhoods: A Mixed-Methods Pilot and Proof of Concept Study"

_ijerph, 2019, doi:10.3390/ijerph16152712_

Reviewer 1 Report

An interesting, well written study looking at a topic that is timely and of great interest to many readers. 

Section 2.3 Line 111: It would be worthwhile including some information related to the length of time that the activities were undertaken by the participants. This could be time for each time the activity was undertaken (ie 1 hour), the number of days/times that an activity was undertaken and the total length of time (such as weeks) that participants were involved in an activity. Do the authors consider that there would be differences in the results if these were undertaken for a longer or shorter time?  

Section 4.1 Line 407: It might be useful to recruit a wider age range of participants and perhaps participants who were not previously involved in the conservation and wildlife groups. Can the authors consider what difference these might make to results? It would be interesting to consider how participation in these activities may influence the health among younger people. 

Author Response

Section 2.3 Line 111: As stated in section 2.5 line 151, all activities lasted 20 to 30 minutes. We have added a note to section 2.5 to state that participants at St Nicks and Barlby undertook activities on one occasion only, whereas as Askham Bog we ran three sessions and nine participants at this site took part in more than one activity. We did not measure dose-response in this study as our primary goal was feasibility and proof of concept. But we state in section 4.2. that it will be important to measure dose response in future evaluations which will have implications on scaling of interventions. We have updated the citation for this point with a recent paper by White et al (Scientific Reports, 2019;9:7730).

Section 4.1 Line 407: We are unable to recruit additional participants to this study which is now complete. We accept that there is scope in future research to evaluate if age moderates the effectiveness of interventions and whether interventions would be efficacious among younger people. An intervention study would include co-design with the target audience to ensure interventions were tailored to meet the needs of specific groups. We have elaborated on this point in the discussion in section 4.2. where we state that it is a priority to undertake definitive research about the mental health benefits of purposeful activity in green space in younger and older adults

Reviewer 2 Report

This study tried to clarify the mental health benefits of purposeful activities in public green spaces with quantitative and qualitative analyses. I can judge the quantitative analysis is fine. However, the qualitative analysis has not enough scientific soundness as an academic paper.

As the author(s) has mentioned in the "4.1 Strength and limitations", the qualitative analysis, in other words, interview survey has conducted to only six people. I will judge it is too small to discuss the research questions which the author(s) has made.

If the author(s) would like to make this manuscript be accepted, the sample size big enough should be collected.

Author Response

Unlike quantitative research, qualitative research rarely seeks, if ever, to offer generalizable findings based on a representative sample. Rather the aim here, as with much qualitative research was on transferability of findings to inform the design of interventions in a definitive controlled study in the future. We acknowledge that the sample size was relatively small (8 people; not 6). However sampling for the qualitative evaluation was based on pragmatic principles and derived on the basis that in feasibility studies sample sizes are usually small, often between 5 and 20 individuals (See O’Cathain et al. Pilot and Feasibility Studies20151:32 https://doi.org/10.1186/s40814-015-0026-y). Further our approach is consistent with the understanding that around 10 users will identify a minimum of 80% of variation in experiences during feasibility testing of interventions (see Faulkner L. Behav Res Methods Instrum Comput. 2003;35(3):379–83.) We have added this additional detail to the strengths and limitations section to further explain our approach to the qualitative component.

Round  2

Reviewer 2 Report

As described in the first review comment, I would still judge the sample size (I am sorry it's eight not six) is not big enough and the conclusion is not highly convincing if I have read this paper.

However, the authors pointed out the limitation in this study clearly. It means the researchers who will read this paper can understand the result with these limitations and will proceed their study considering this study.

In conclusion, I will judge this study can be accepted and published.